# Similarity of introduced plant species to native ones facilitates naturalization, but differences enhance invasion success

Jan Divíšek [1,2], Milan Chytrý [1], Brian Beckage [3], Nicholas J. Gotelli [4], Zdeňka Lososová [1], Petr Pyšek [5,6], David M. Richardson [7] & Jane Molofsky [3]

The search for traits associated with plant invasiveness has yielded contradictory results, in part because most previous studies have failed to recognize that different traits are important at different stages along the introduction–naturalization–invasion continuum. Here we show that across six different habitat types in temperate Central Europe, naturalized non-invasive species are functionally similar to native species occurring in the same habitat type, but invasive species are different as they occupy the edge of the plant functional trait space represented in each habitat. This pattern was driven mainly by the greater average height of invasive species. These results suggest that the primary determinant of successful establishment of alien species in resident plant communities is environmental filtering, which is expressed in similar trait distributions. However, to become invasive, established alien species need to be different enough to occupy novel niche space, i.e. the edge of trait space.

---

[1] Department of Botany and Zoology, Masaryk University, Kotlářská 2, 611 37 Brno, Czech Republic. [2] Department of Geography, Masaryk University, Kotlářská 2, 611 37 Brno, Czech Republic. [3] Department of Plant Biology, University of Vermont, Burlington, VT 05405, USA. [4] Department of Biology, University of Vermont, Burlington, VT 05405, USA. [5] Department of Invasion Ecology, Institute of Botany, The Czech Academy of Sciences, 252 43 Průhonice, Czech Republic. [6] Faculty of Science, Department of Ecology, Charles University, 128 43 Praha 2, Czech Republic. [7] Centre for Invasion Biology, Department of Botany and Zoology, Stellenbosch University, Matieland 7602, South Africa. Correspondence and requests for materials should be addressed to J.M. (email: Jane.Molofsky@uvm.edu)

**B**iological invasions threaten the world's biodiversity and the functioning of ecosystems. Over 13,000 plant species have successfully naturalized around the world[1,2]; some of them have become invasive and there is no sign that their numbers will decrease in the near future[3]. Therefore, policy and management decisions require predictive tools to better assess the likelihood of individual introduced species becoming invasive[4].

Functional trait analyses have become an increasingly powerful tool to determine the constraints placed on plant species by the environment[5,6]. The question of plant invasiveness, i.e., why some plant species invade areas outside their native range more than others, has been addressed in several studies that compared traits of native vs. alien species[7,8] or non-invasive vs. invasive species[9,10]. There are two contrasting hypotheses that predict the outcome of such comparisons. First, the environmental filtering hypothesis[11,12] postulates that alien species need to be similar to native species occurring in the same habitat because a set of specific traits enables certain species, both native and alien, to establish and persist in a certain habitat. Second, the limiting similarity hypothesis[13,14] suggests that alien species need to be different from native species to avoid niche overlap and competition with resident species in the community or to be able to outcompete resident species. These hypotheses also suggest that invasion processes are driven by an interplay between the phylogenetic position of the invading species and the phylogenetic structure of the invaded community. The environmental filtering hypothesis assumes that phylogenetic relatedness of invaders to native species promotes naturalization because phylogenetically related alien species tend to have similar functional traits and environmental adaptations as native species[15,16]. In contrast, under the limiting similarity hypothesis, phylogenetic relatedness can hamper naturalization because of the stronger competition of aliens with native species[17,18].

Although comparative studies of traits of native and alien species have been reviewed and synthesized[19,20] or subject to meta-analyses[8,9,21], the results remain equivocal. Hulme and Bernard-Verdier[22] summarize the main causes of the current lack of consensus; they emphasize the context-dependence of individual studies, the role of factors such as different spatial scales studied, and the choice of different traits.

An important issue that should be tackled within trait-based studies of invasiveness is the position of alien species along the introduction–naturalization–invasion (INI) continuum[23,24]. Specifically, naturalized non-invasive species, i.e., alien species that are reproducing in the wild but not spreading fast in the introduced range, may have different traits than invasive species, i.e., alien species that spread over considerable distances[25]. There is evidence that traits of non-invasive species tend to be different from those of invasive species[10,26,27], and that differences in traits between native and invasive species are larger than the differences between native and naturalized non-invasive species[9]. Therefore, studies that do not differentiate between naturalized non-invasive and invasive species obscure crucial differences between less successful and successful aliens. Such studies provide spurious information on the functional causes of invasion success and hinder efforts to reduce the impacts of invasions.

Another important issue in the trait-based study of species invasiveness is the habitat context of invasions. Different habitats are invaded by different plant species[28], whereas alien species pools are composed of subsets of species adapted to many different habitats[29,30]. The traits that allow species to successfully establish in certain habitats and persist in plant communities associated with such habitats may not be advantageous in another habitat. Studies that address the habitat/community context appropriately are mainly done at fine spatial scales, but insights from such studies are difficult to generalize. Broad-scale synthetic studies usually replace habitat types by proxies such as landscape sections or biomes[8,21], which are still quite heterogeneous, potentially comprising several, often contrasting, habitat types. Therefore, trait comparisons of native and invasive species should be done within each habitat for which native and invasive species are already competing. Comparisons within and across multiple habitat types, accurately identified in the field and documented by representative sampling across large areas, are necessary to reveal traits that allow alien species to invade in different conditions and compete with different native species.

Here we compare the functional traits of native and naturalized alien species (the latter category being subdivided into naturalized non-invasive, and invasive species following the criteria proposed by Richardson et al.[23] and Blackburn et al.[24]), in the context of habitat types. Our comprehensive dataset consists of 24,935 vegetation plots (phytosociological relevés) from the Czech Republic in which 1438 native, 261 naturalized non-invasive and 50 invasive species were recorded. Based on this dataset, we compiled lists of species occurring in each of the six broadly defined habitats of this country (Table 1). For each species, we identified three functional traits, namely the specific leaf area (SLA), maximum plant height and seed weight. We selected these three traits because they capture a large part of the ecologically significant differences among species[31] and they are easily measurable and available in species trait databases[32,33]. SLA represents how fast a species can acquire resources, maximum plant height represents how well a species performs in competition, and seed weight represents species position on the r–K continuum[34], in which there is a tradeoff between quantity and quality of offspring: r-strategists produce many offspring, each of which has a relatively low probability of surviving to adulthood, whereas K-strategists invest more heavily in fewer offspring, each of which has a relatively high probability of surviving to adulthood. The comparisons were performed for both observed functional traits, which tend to be similar among closely related species due to shared phylogenetic history[35,36] and residuals of phylogenetic models in which species' phylogenetic non-independence was accounted for.

We show that across six different habitat types in temperate Central Europe, naturalized non-invasive species (i.e., those that establish and reproduce but do not spread considerably) are functionally similar to native species while invasive species (i.e. those that spread over considerable distances from introduction sites) are functionally different. This pattern was driven mainly by the greater average height of invasive species and did not qualitatively change after we statistically controlled for the effects of phylogenetic relatedness.

## Results

**Single-trait comparisons.** For each habitat type, we compared traits of naturalized and invasive species with those of native species and applied simple randomization tests of the null hypothesis that univariate trait median of either naturalized or invasive species is not significantly different from that of native species. Single-trait comparisons between native and naturalized species yielded only two significant results ($p \leq 0.05$) when the correction for multiple testing was applied (Fig. 1, Supplementary Table 1): naturalized non-invasive species in grasslands and forests were respectively 0.2 m and 0.3 m taller than native species in these habitats. In contrast, comparisons of invasive and native species revealed that invasive species were on average 1.2 m taller than native species across all habitats and also had 2.6 mg heavier seeds but only in forest vegetation. For SLA, we found no differences between native and invasive species in any habitat.

**Table 1 Numbers of species and their average percentage covers in six habitat types**

| Habitat type | Native species | Naturalized non-invasive species | Invasive species |
|---|---|---|---|
| Grassland and heathland vegetation | 1161 (6.3) | 160 (3.3) | 29 (8.8) |
| Ruderal and weed vegetation | 841 (6.9) | 224 (6.3) | 39 (11.8) |
| Rock and scree vegetation | 344 (3.5) | 68 (8.3) | 11 (3.5) |
| Wetland vegetation | 636 (22.4) | 101 (16.7) | 28 (3.9) |
| Scrub vegetation | 774 (9.8) | 109 (3.3) | 24 (10.4) |
| Forest vegetation | 957 (9.1) | 102 (4.1) | 38 (16.9) |
| Total | 1438 (11.2) | 261 (6.5) | 50 (10.7) |

Average percentage covers shown in parentheses were calculated as the arithmetic means of species covers in vegetation plots classified to each habitat type. Note that tree species occurring in the first four habitat types (usually as juvenile individuals) were removed

As traits of evolutionarily closely related species were more similar than expected at random (according to the Abouheif's $C_{mean}$ statistics), we repeated the above-mentioned univariate comparisons for residuals of phylogenetic models to account for phylogenetic non-independence of species. These comparisons only slightly altered our previous results (Table 2, Supplementary Table 1, and Supplementary Figure 1); we found that naturalized species were not significantly different from native species in any of the habitats. For invasive species, the difference found in phylogenetically non-informed comparison was retained, with invasive plants being significantly taller in all habitats except rock and scree vegetation.

The analyses were repeated for a dataset where missing trait values were imputed based on correlations among traits and species' phylogenetic relatedness, yielding almost the same results (Supplementary Table 6, and Supplementary Figures 2 and 3).

**Combined trait comparisons**. To explore the distribution of native, naturalized, and invasive species in the trait space of each habitat, we first plotted the location of each species in three-dimensional space with axes defined by $log_{10}$-transformed and standardized SLA, species height, and seed weight (Fig. 2, Supplementary Figures 4–7). Next, we defined the centroid of each trait space as the arithmetic mean of native species traits and then divided the trait space into eight regions (octants) based on the eight possible combinations of above-average or below-average SLA, species height, and seed weight. We then calculated the proportions of native, naturalized non-invasive, and invasive species in these octants. We found that naturalized non-invasive species occupy almost all octants of the trait space in relatively similar proportions (Fig. 3, Supplementary Table 3). In contrast, invasive species occurred more frequently in octants with above-average plant height and seed weight. This pattern did not change after accounting for phylogenetic relatedness of species (Supplementary Figure 8 and Supplementary Table 4). The dataset with imputed trait values showed very similar patterns (Supplementary Figures 9–10 and Supplementary Tables 5–6).

We then measured the distance of each species from the native group centroid in the trait space of each habitat (Supplementary Figure 11). Average distances of native species ranged from 1.25 to 1.46 SD units (25–31% of the most distant species), those of naturalized species from 1.11 to 1.36 SD (23–27%), and those of invasive species from 1.60 to 2.11 SD (36–43%). Across all the habitats, naturalized species were thus on average 0.07–0.18 SD (1–4%) closer to the centroid than native species. This result indicates high functional similarity of traits of naturalized and native species. In contrast, invasive species were on average 0.30–0.67 SD (7–15%) further than native species from the group centroid, indicating their higher functional dissimilarity compared

with average native species. This pattern was retained when we analyzed residuals of phylogenetic models, although average distances from the centroid were shorter for each species group (Supplementary Figure 12). The dataset with imputed trait values showed very similar patterns (Supplementary Figures 13–14).

To test whether the distribution of naturalized and invasive species in the trait space differs significantly from the distribution of native species, we constructed for each species group the cumulative distribution function (CDF) of species' distances from the native group centroid of the trait space. We applied a randomization test to decide whether the CDF for alien species (either naturalized or invasive) is significantly above or below the CDF of native species (i.e., that alien species are not significantly closer or further from the native group centroid than native species). Across all the habitats, the CDFs for naturalized non-invasive species were above the CDFs for native species, indicating that naturalized species have trait values that are consistently close to the average traits of native species (Fig. 4, Supplementary Table 7). However, randomization tests (followed by the correction of resulting $p$-values for multiple testing) revealed that differences between the CDFs for naturalized and native species were not statistically significant ($p > 0.05$) in five of six habitats. In grassland and heathland vegetation, the CDF for naturalized species was significantly above the CDF for native species, indicating that traits of naturalized species were more similar to native species than expected for a random group of species. In contrast, the CDFs for invasive species were always below the CDFs for native species (Fig. 4, Supplementary Table 7), indicating that invasive species occupy the periphery of the trait space in each habitat. Randomization tests revealed that these differences were statistically significant ($p \leq 0.05$ after the correction for multiple testing) in all habitat types except wetland ($p_{adj.} = 0.064$).

When residuals from the phylogenetic models were analyzed, the CDFs for naturalized species were still above the CDFs for native species in all habitats and there were no significant differences between the two species groups (Table 3, Supplementary Table 7, and Supplementary Figure 15). Phylogenetic analyses of native vs. invasive species did not alter the qualitative pattern but did alter significance values. Across the six habitats, the invasive species CDFs were always below the native species CDFs, but the differences were not statistically significant ($p > 0.05$ after the correction for multiple testing).

The dataset with imputed trait values showed very similar patterns. Across all habitats, the naturalized non-invasive species were statistically indistinguishable from native species, whereas invasive species were significantly further from the average trait values of native species (Supplementary Table 8 and Supplementary Figure 16). When residuals of phylogenetic models were analysed, the differences between CDFs were not statistically

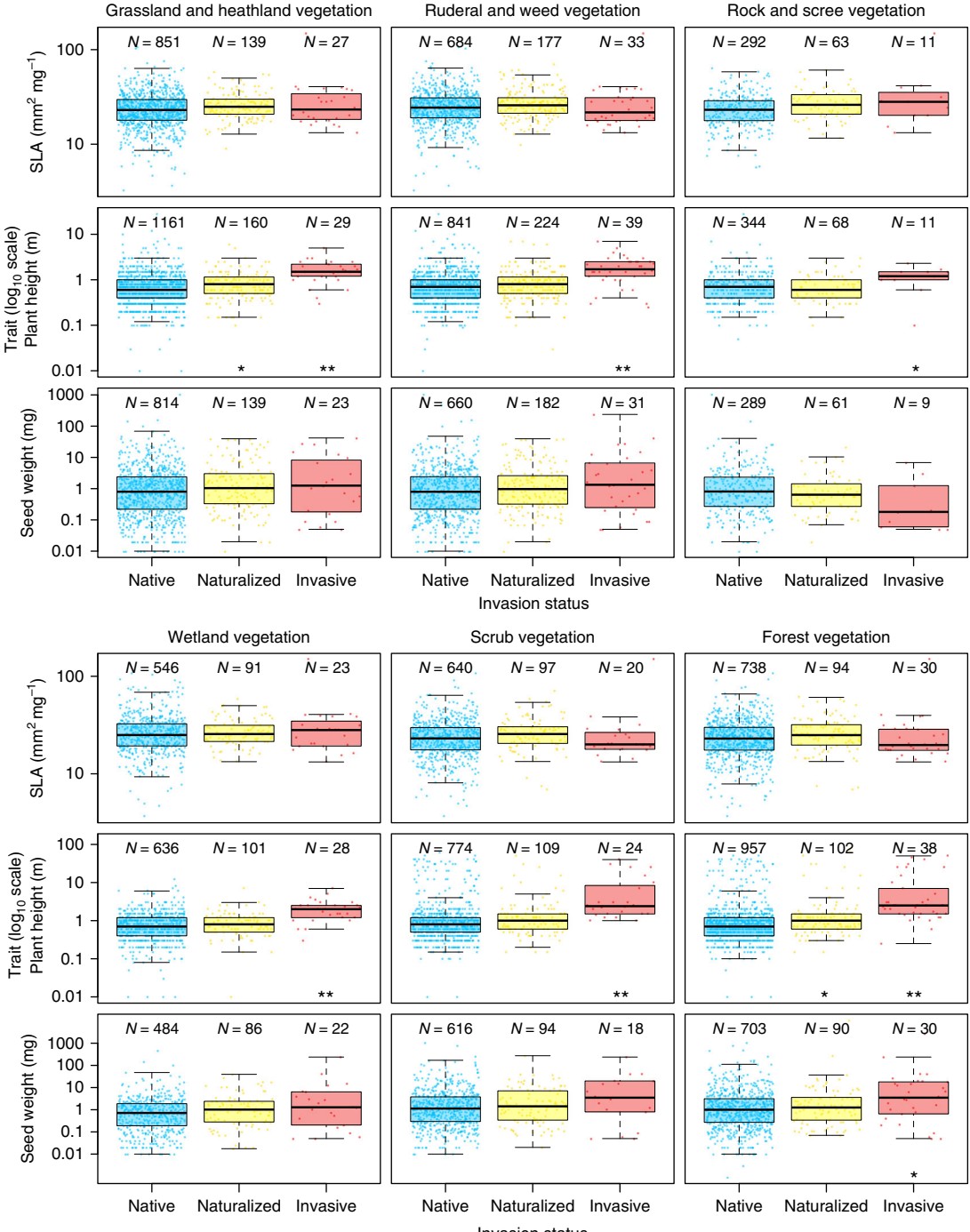

**Fig. 1** Distribution of species traits (log$_{10}$ scale) in each habitat type. The number of native, naturalized non-invasive and invasive species (N) is indicated above each boxplot (species with missing trait values were removed). Asterisks below boxplots for naturalized and invasive species indicate statistical significance of their difference from native species established using randomization tests and adjusted using Benjamini and Hochberg's correction method:[55] ***$p \leq 0.001$; **$0.001 < p \leq 0.01$; *$0.01 < p \leq 0.05$. For complete results of randomization tests, see Supplementary Table 1. The thick horizontal line in each box indicates the median. The bottom and top of each box indicate the 25th and 75th percentiles, respectively, and the vertical lines (whiskers) represent either the maximum/minimum value or 1.5 × interquartile range depending on which is closer to the mean. Outliers are indicated by jittered points outside the range of whiskers

significant for both naturalized and invasive species (Supplementary Table 8 and Supplementary Figure 17).

## Discussion
By distinguishing between different stages of the INI continuum, we can disentangle the two competing hypotheses: introduced species must share some characteristics to enter the community (i.e., environmental filtering hypothesis[11,12]) but to disrupt the community introduced species must be dissimilar in traits (i.e., the limiting similarity hypothesis[13,14]). We found that in a temperate flora of Central Europe, traits of naturalized non-invasive species are similar to, whereas those of invasive species are dissimilar from, the traits of native species. This pattern was

**Table 2 Differences (Δ) of median trait values between alien and native species after accounting for phylogenetic relationships among species**

|  | Δ SLA | $p_{adj.}$ | Δ Plant height | $p_{adj.}$ | Δ Seed weight | $p_{adj.}$ |
|---|---|---|---|---|---|---|
| **Naturalized non-invasive vs. native species** |  |  |  |  |  |  |
| Grassland and heathland vegetation | 0.013 | 0.688 | 0.008 | 0.626 | 0.026 | 0.824 |
| Ruderal and weed vegetation | 0.008 | 0.688 | − 0.023 | 0.486 | 0.018 | 0.824 |
| Rock and scree vegetation | 0.004 | 0.688 | − 0.027 | 0.486 | 0.025 | 0.824 |
| Wetland vegetation | − 0.011 | 0.688 | − 0.038 | 0.486 | 0.039 | 0.824 |
| Scrub vegetation | 0.024 | 0.688 | 0.012 | 0.626 | 0.035 | 0.824 |
| Forest vegetation | 0.006 | 0.688 | 0.024 | 0.486 | 0.006 | 0.942 |
| **Invasive vs. native species** |  |  |  |  |  |  |
| Grassland and heathland vegetation | − 0.034 | 0.299 | **0.128** | **0.007** | − 0.069 | 0.425 |
| Ruderal and weed vegetation | − 0.048 | 0.256 | **0.152** | **0.003** | − 0.061 | 0.425 |
| Rock and scree vegetation | 0.030 | 0.521 | 0.120 | 0.094 | − 0.275 | 0.132 |
| Wetland vegetation | − 0.053 | 0.256 | **0.148** | **0.006** | − 0.053 | 0.546 |
| Scrub vegetation | − 0.050 | 0.256 | **0.146** | **0.007** | − 0.077 | 0.425 |
| Forest vegetation | − 0.044 | 0.256 | **0.158** | **0.003** | 0.103 | 0.425 |

$p_{adj.}$ is a probability value resulting from the randomization test adjusted using Benjamini and Hochberg's correction method[55]. **Differences that were significant after this correction are in bold. For complete results of randomization tests see Supplementary Table 1**

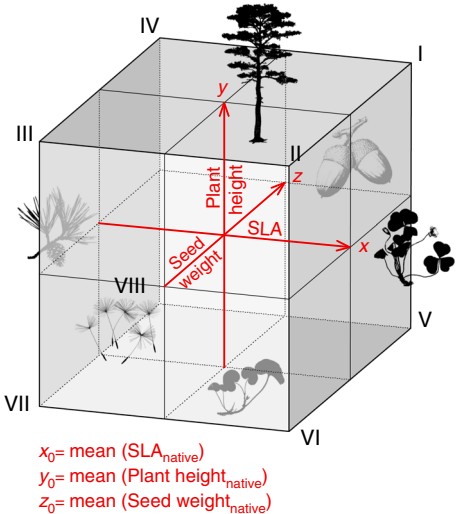

$x_0$= mean (SLA$_{native}$)
$y_0$= mean (Plant height$_{native}$)
$z_0$= mean (Seed weight$_{native}$)

**Fig. 2** A trait space divided into eight regions (octants). The centroid of the trait space was defined as the arithmetic mean of SLA (x), plant height (y), and seed weight (z) of native species occurring in the habitat type. Octants I–IV include species with above-average height, whereas octants V–VIII include species with below-average height. Analogously, octants I, II, V, and VI include species with above-average SLA, whereas octants III, IV, VII, and VIII include species with below-average SLA. Finally, species with above-average seed weight occupy octants I, IV, V, and VIII, whereas species with below-average seed weight occupy octants II, III, VI, and VII. Drawings created by J. Divíšek or redrawn from photographs by Pavel Veselý (*Oxalis* and *Asarum*) with kind permission

consistent across the six studied habitat types. Functional similarity to native species is sufficient for successful naturalization of introduced species. However, to become invasive, alien species need to be functionally different from the mean trait values of native species. In other words, a species needs to be similar enough to be admissible to a community type, but different enough, i.e., situated on the edge of the trait space, to become invasive. Our analysis also indicates that in all habitat types, the trait that makes this difference is being taller at maturity, which suggests that stronger competitive ability is the key to success of invasive alien species in this flora[37]. We expect the location of

invasive species to be on the edge of trait space in other floras too, although the nature of the traits may be different.

A global meta-analysis by van Kleunen et al.[9] showed that invasive species had on average higher values of several traits than non-invasive species and more trait differences were significant for native vs. invasive comparison than for non-invasive alien vs. invasive comparisons. Gallagher et al.[10] found that in the Australian flora, traits of invasive plants differed from those of naturalized non-invasive congeneric species. However, unlike our study, none of these analyses were based on the total flora of individual habitats.

The similarity of non-invasive alien and native species suggests that environmental filtering is the main mechanism that controls establishment of newly introduced species into plant communities in invaded habitats. Environmental filtering[11,12] assumes that new species require the same conditions as native species, which is a common pattern among naturalized seed plants. If naturalized species have the same values of key traits as resident native species, the establishment of an individual of a new species in the community may be equally probable as the establishment of an individual of a resident species, assuming equal arrival of seeds into the habitat. Such new species can successfully become integrated into the community, but they do not expand.

The ability to expand, spread over long distances or become dominant in communities, i.e., to become invasive[23], requires that species possess traits or trait combinations that are novel for the community (phenotypic divergence[8]). In this study, we found that invasive species tend to occupy the edge of the trait space in each habitat, suggesting that they have unique combinations of the three considered traits that allow them to become invasive. This methodology suggests that an analysis of trait space could lead to a prediction of the likely invasiveness of an introduced species—it will be maximized near the edge of multivariate trait space and decline with either increasing distance toward the centroid or away from the edge. Towards the centroid, the species will become too similar to the native species (limited by competition with native species) and thus will likely become naturalized but not invasive, while further away from the centroid, it will become too different from the native species (and will be excluded by environmental filtering).

In our study, the difference from the average traits of native species was driven mainly by plant height and less so by seed weight and SLA. This suggests that being taller and able to outcompete other species through shading is the key to the success of

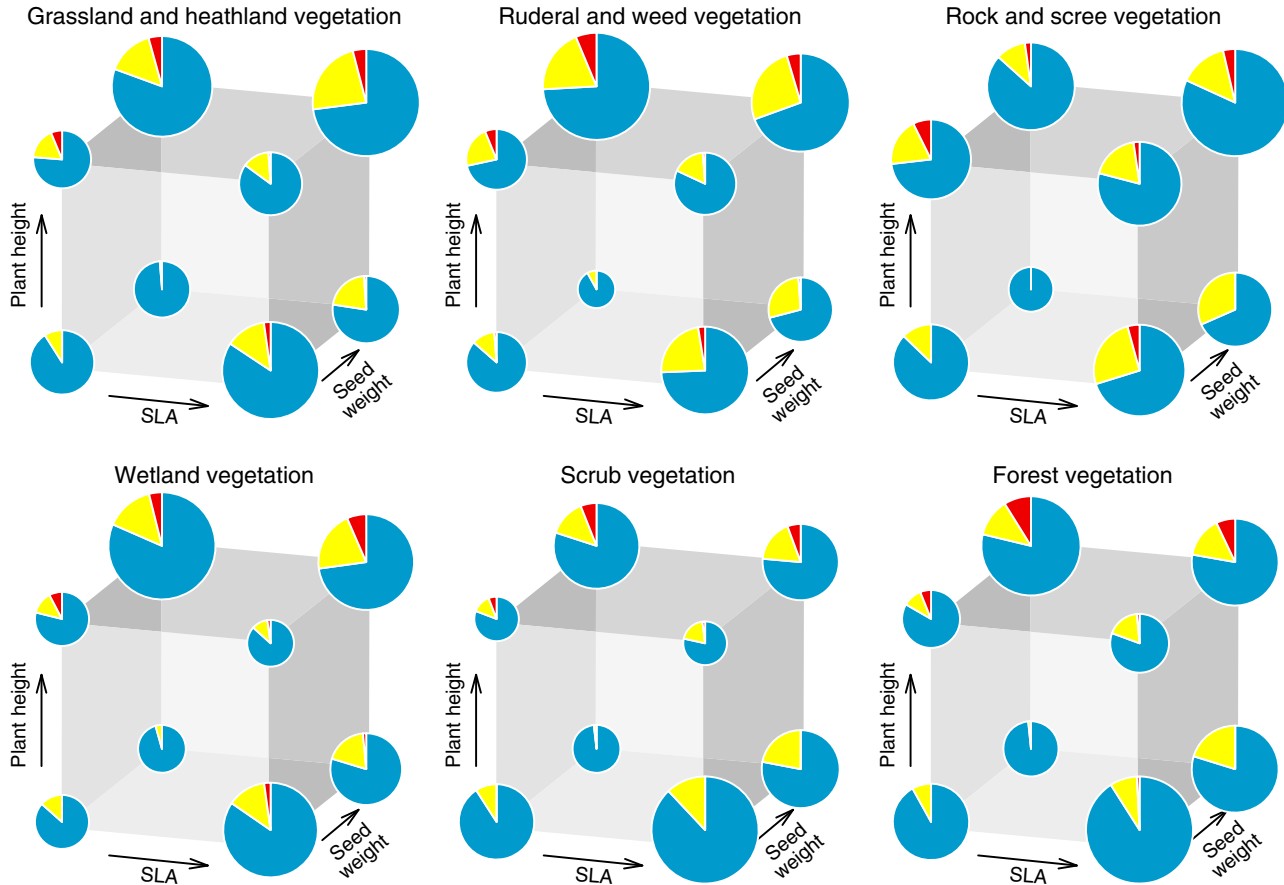

**Fig. 3** Proportion of species occupying octants of the trait space in each habitat. Blue, yellow, and red colors represent, in turn, native, naturalized non-invasive, and invasive species. Octants were defined with respect to the centroid of the native species group in the trait space, i.e., by above-average or below-average SLA, plant height, and seed weight of native species (Fig. 2). The bigger the pie, the larger number of species occupies the region. For numbers of species in each octant, see Supplementary Table 3

invasive species in the temperate ecosystems of Central Europe[38]. In a study comparing native species with aliens (not considering whether species were invasive or only naturalized), Ordonez et al.[8] found that aliens were actually on average shorter than natives, but this difference disappeared when the comparisons were made within individual growth forms. Our analysis compared traits within habitat types while excluding occasional occurrences of (mostly juvenile) trees from open habitat types; therefore, we also largely compared species within the same or similar growth forms. The importance of height in our results therefore does not reflect invasion of alien trees into treeless habitats (where multiple mechanisms mediate establishment and invasions[39]), but invasions of taller species into communities of native species dominated by a growth form shared with the invading alien.

Our results contradict those of several previous studies that reported larger SLA for alien or invasive species[8,10,26,40]. SLA is a leaf trait indicative of fast growth rate[41,42], a property that can be advantageous for invasion. However, most of the previously analyzed datasets on plant traits were from floras dominated by woody plants. In the floras dominated by herbaceous plants with generally high SLA, such as the Central European flora analyzed here, the trait relationships can be different[43,44]. Effects of SLA and associated traits such as growth rate on plant invasiveness in herb-dominated floras clearly require further study.

We found some evidence for larger seeds of invasive species in forests and the difference between invasive and native species in seed size was marginally significant in scrub vegetation. This is in contrast with other studies showing that invasiveness is correlated with small seed size[8,26,45], which we found only for rock and scree vegetation and residuals of phylogenetic models but these differences were not statistically significant. Our result may be attributable to the positive correlation of seed mass with plant height[6]. If so, then invasion success in the herb-dominated flora of a temperate biome would have a simple explanation, involving a single trait and single mechanism rather than a complex life-history syndrome involving interactive effects of many different traits. However, it is important to keep in mind that this explanation is valid if trait effects are considered within the context of individual habitats. Also, it should be noted that our study considered regional species pools of individual habitats rather than local plant communities within these habitats, and that trait effects can vary across scales[22].

The environmental filtering hypothesis assumes phylogenetic conservatism of ecological adaptations because alien species that are closely related to the resident native species most likely have functional traits advantageous for successful naturalization. Conversely, the limiting similarity hypothesis postulates that alien species that have close relatives in invaded habitats would be less successful due to high overlap in traits and therefore competition for similar resources with resident native species. Accounting for phylogenetic signal in species traits may thus help to disentangle the role of phylogeny in the invasion process. In our study, results obtained after controlling for the phylogenetic signal did not alter our main conclusions. Observed patterns were retained with naturalized species being very close or indistinguishable from

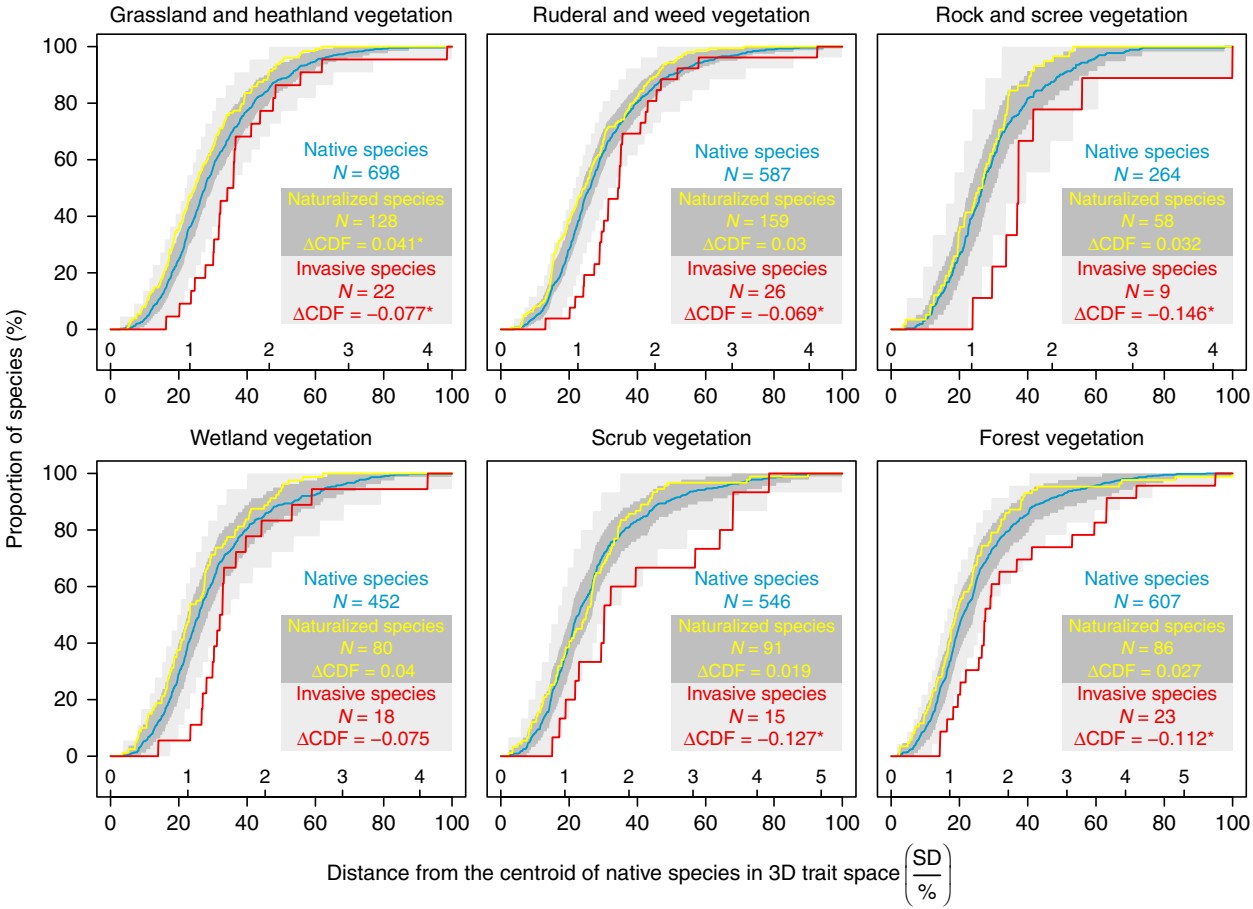

**Fig. 4** Species distribution in the trait space of each habitat. Cumulative distribution functions (CDF) show the cumulative number of species at each distance from the centroid of the native species in the three-dimensional trait space of each habitat type. The horizontal axis shows both the original distances (SD units; above the axis) and relative distances (%, below the axis). Blue, yellow, and red lines represent, in turn, native, naturalized non-invasive, and invasive species. The dark grey area shows a 95% confidence interval of simulated CDFs for naturalized species, whereas the light grey area shows a 95% confidence interval for invasive species. ΔCDF is the observed area between the curves calculated after scaling species distances in each habitat to relative values. Positive values indicate that the CDF for alien species (either naturalized non-invasive or invasive species) or its prevalent part is above the CDF for native species (i.e., traits of alien species are similar to the average trait of native species), whereas negative values indicate that the CDF for alien species is below the CDF for native species (i.e., traits of alien species are dissimilar to the average trait of native species). Statistical significance of the difference between the CDFs for alien and native species resulting from randomization test and adjusted using Benjamini and Hochberg's correction method[55] is indicated by asterisks: ***$p \leq 0.001$; **$0.001 < p \leq 0.01$; *$0.01 < p \leq 0.05$. Species with a missing value of any of the three considered traits were removed. For complete results of randomization tests, see Supplementary Table 7

---

**Table 3 Differences between the cumulative distribution functions (CDFs) of alien and native species after accounting for phylogenetic relationships among species**

|  | ΔCDF for naturalized species | $p_{adj.}$ | ΔCDF for invasive species | $p_{adj.}$ |
|---|---|---|---|---|
| Grassland and heathland vegetation | 0.026 | 0.246 | − 0.059 | 0.124 |
| Ruderal and weed vegetation | 0.019 | 0.344 | − 0.082 | 0.124 |
| Rock and scree vegetation | 0.028 | 0.344 | − 0.091 | 0.153 |
| Wetland vegetation | 0.034 | 0.246 | − 0.065 | 0.124 |
| Scrub vegetation | 0.018 | 0.344 | − 0.063 | 0.124 |
| Forest vegetation | 0.016 | 0.362 | − 0.048 | 0.127 |

ΔCDF for naturalized species is the observed area between the CDFs for native and naturalized non-invasive species. ΔCDF for invasive species is the observed area between the CDFs for native and invasive species. Positive values indicate that the CDF for alien species or its prevalent part is above the CDF for native species (i.e., traits of alien species are similar to the average trait of native species), whereas negative values indicate that the CDF for alien species is below the CDF for native species (i.e., traits of alien species are similar to the average trait of native species). $p_{adj.}$ is a probability value resulting from the randomization test adjusted using Benjamini and Hochberg's correction method[55]. For complete results of randomization tests see Supplementary Table 7

average trait values of the native community and invasive species being further from this average, but the statistical significance of this difference was reduced. Considering that differences in plant height between native and invasive species remained significant (when analyzed separately) even after accounting for species phylogenetic relatedness, the decrease of statistical significance of the difference between CDFs could be ascribed to reduced correlation among traits. It has been shown in other studies that alien species tend to invade communities with close relatives because of their shared adaptations to the same environments, which support successful naturalization[15,16]. However, the excessive similarity of alien species to native ones might also prevent naturalized species from becoming invasive because they lack competitive advantage. Therefore, a small subset of the naturalized species that differ from the native species of the same habitat in possessing additional traits that support competitiveness, namely taller stature, can eventually become invasive. The fact that this pattern persists even in residuals of phylogenetic models suggests that it is turning up repeatedly for species from different lineages.

We conclude that naturalized non-invasive species tend to be functionally similar to native species occurring in the same habitat type, but that invasive species differ from both native and naturalized non-invasive species by occupying the periphery of the plant functional trait space represented in each habitat, i.e., the edge of the trait space. We speculate that, for the temperate herb-dominated flora of Central Europe, this functional difference can be largely ascribed to a single easily measurable and widely available trait, plant height. Additional height gives alien species better access to light that may enhance their competitive ability and allow them to achieve invasive status. However, we expect the distinguishing traits to be flora specific. Although the edge of the trait space result is not expected to be specific to any community, it suggests that the probability of an introduced species becoming invasive can be predicted a priori. With the advent of global plant functional trait databases such as TRY[33], the approach outlined here may be used as a screening tool for determining which introduced plants have the highest probability of becoming invasive in the future.

## Methods

**Data.** The data were derived from records of plant species composition in 24,935 vegetation plots sampled across the Czech Republic, Central Europe, obtained from the Czech National Phytosociological Database[46] (GIVD code EU-CZ-001). This dataset provides the most comprehensive information on plant species composition of major habitat types in Central Europe. There is also very good knowledge of the status of all alien plant species recorded in these habitats. The size of vegetation plots in the dataset ranged from 1 to 625 m², being proportional to the size of dominant plants. Based on its species composition, each plot record was assigned using a computer-based expert system[47] to one of six mutually exclusive habitat types (Table 1): (1) Grassland and heathland vegetation below the timberline (6554 plots), (2) Ruderal and weed vegetation (6265 plots), (3) Rock and scree vegetation (335 plots), (4) Wetland vegetation (6378 plots), (5) Scrub vegetation (553 plots), and (6) Forest vegetation (4850 plots). If juvenile trees occurred in the plots of the first four habitat types, they were removed because their functional traits (see below) are related to fully grown individuals. This dataset contained a total of 1855 seed plant species (all non-flowering plants were removed), of which 417 were classified as alien to the Czech Republic, i.e., as species present in the country because human actions enabled them to overcome fundamental biogeographical barriers (human-mediated extra-range dispersal); they occur in the area as a result of intentional or accidental introduction by humans, or due to spontaneous spread from other areas where they were introduced by humans[48]. Alien species were further subdivided according to the most advanced stage they are known to have reached in the Czech Republic along the INI continuum that describes how species proceed in the invasion process by overcoming geographical, environmental, and biotic barriers[23,24,49]. Based on this concept, we classified alien species as casual, naturalized non-invasive (hereafter naturalized), or invasive (Table 1). Casual species were defined as alien species that do not form self-sustaining populations in the invaded region. However, as they are less widely distributed than naturalized species, they rarely occurred in vegetation plots and we excluded them from the analyses (106 species). Naturalized non-invasive species were defined as alien

species that form self-sustaining populations for several life cycles without direct intervention by people, often recruiting offspring freely, usually close to adult plants; their persistence does not depend on the ongoing input of propagules. Finally, invasive species were defined as alien species that form self-replacing populations over many life cycles, produce reproductive offspring, often in very large numbers at considerable distances from the parent and/or site of introduction, and have the potential to spread over long distances. For a detailed classification of alien species of the Czech flora, see Pyšek et al.[48]. Based on this dataset, we compiled a list of species occurring in each habitat type (see Supplementary Data 1).

For each species, measures of three functional traits were extracted from published literature[50] and the LEDA database:[32] (1) SLA (mm² mg⁻¹), (2) maximum plant height (m), and (3) germinule (hereafter seed) weight (mg). This choice of traits follows the proposal of Westoby[31], assuming that these three traits capture a large part of the ecologically significant differences among species: SLA is related to how fast can species capture light resources with a larger SLA, indicating a greater efficiency in light capture per unit biomass; height can be thought of as a surrogate for a plant's competitive ability as taller plants can generally outcompete smaller ones; and seed weight represents species position on the r–K continuum[34], as species with smaller seeds usually reproduce quickly with many offspring, whereas those with larger seeds are more likely to survive to adulthood.

**Imputation of missing trait values.** As SLA was not available for 29% of species and seed weight for 33% of species (Supplementary Table 9), and common practice of removing missing data not only reduces sample size but may also introduce bias that can lead to incorrect conclusions[51], we prepared three alternative datasets in which missing trait values were imputed using three different methods. First, we imputed missing traits by simple averaging of available trait values across genera or families if trait values for all species of the genus were missing. Second, we used the Random Forest algorithm (with default settings) implemented in the R package missForest[52] to impute missing trait values based on relationships among available traits. Third, we followed the recommendation of Penone et al.[51] and also included phylogenetic information in the form of the first 10 phylogenetic eigenvectors obtained from phylogenetic eigenvector analysis[53,54] (for details see below) as additional predictor variables in the Random Forest model. As all of these imputation approaches gave qualitatively very similar results in subsequent analyses, we show only the results based on (i) original dataset where species with missing trait values were removed; and (ii) the last dataset in which missing trait values were imputed based on correlations among traits and information of phylogenetic relatedness.

**Single-trait analysis.** As the basic units of the analyses were individual habitat types (not vegetation plots), the analyses address regional species pools of individual habitats (not local plant communities). We first removed species with missing trait values and compared traits of native, naturalized, and invasive species occurring in each habitat type using boxplots. We applied simple randomizations to test the null hypothesis that univariate trait medians of either naturalized or invasive species were not significantly different ($p < 0.05$, two-tailed test) from the median of native species. In each of the 999 randomizations, trait values were randomly re-shuffled between native and naturalized species or between native and invasive species. Resulting $p$-values were then adjusted using Benjamini and Hochberg's[55] method to avoid issues connected with multiple testing. This analysis was also repeated with a dataset where missing trait values were imputed based on correlations among traits and information about species phylogenetic relatedness to ensure that removing species with missing trait values did not affect our results.

**Multiple-trait analysis.** To explore the distribution of native, naturalized and invasive species in the trait space occupied in each habitat, we first $\log_{10}$-transformed (to reduce the effect of extreme trait values and skewness of the data) and scaled each trait to zero mean and unit variance (z-transformation). Species with a missing value of any of the three traits were removed (see Supplementary Table 9 for numbers of species with available values of all three traits). We then plotted the location of each species in a three-dimensional trait space with axes defined by SLA, plant height, and seed weight, and calculated the location of the unweighted group centroid of the native species in the trait space of each habitat (i.e., the arithmetic mean of traits of native species occurring in the habitat). Based on this centroid, we divided the trait space into eight regions (octants), each one defined by one of the eight possible combinations of above-average or below-average values of SLA, plant height, and seed weight (Fig. 2). We then calculated the proportion of native, naturalized non-invasive and invasive species in these regions, and also measured the distance of each species from the native group centroid to determine whether any group of species is distributed further from the centroid than others. For these kinds of data, a parametric $t$-test could be used to compare the average distance of each species group (native, naturalized, and invasive) from the centroid. However, this kind of test compares only the means of the two groups, whereas there could be differences in the variance, skewness, or other moments of the distribution. Parametric tests are available for higher moments, but these tests are

sensitive to outliers and influential points. Therefore, we used a more general test of the difference in the CDFs for native species and naturalized or invasive species. A standard Kolmogorov–Smirnov CDF test is based on the maximum difference between two curves and a look-up table is used to estimate a tail probability for CDF differences based on ranked observations[56]. For our test, we first scaled species distances in each habitat to relative values and then calculated the summed area difference between the CDF for native species and the CDF for either naturalized or invasive species (ΔCDF; see also Cayuela et al.[57] for a similar approach to comparing rarefaction curves). In approx. 50% of the comparisons, the two CDFs crossed one another. When this happened, we used the larger absolute value of the differences calculated in the two regions that were created by the crossing. This larger part represented on average about 96% of the total area between the curves. To test the null hypothesis that two CDFs are not significantly different ($p < 0.05$, two-tailed test), we randomly re-shuffled (999 randomizations) trait values between the native and invasive species or between the native and naturalized species, and constructed a set of 999 simulated CDFs. If simulated CDF crossed observed CDF for native species, the same procedure as mentioned above was used to estimate the area difference. To account for multiple testing, we adjusted the resulting p-values using Benjamin and Hochberg's[55] method. Finally, this analysis was also applied to the dataset with imputed trait values.

**Phylogenetic signal in the single-trait analysis**. Traits of evolutionarily closely related species tend to be more similar than expected at random[35,36] and this phylogenetic autocorrelation is often called phylogenetic signal[58]. The functional similarity of alien species to native ones caused by shared evolutionary history may promote their successful naturalization[15,16] but, on the other hand, prevent them from becoming invasive[17,18]. We measured the similarity of species' trait values using Abouheif's $C_{mean}$ statistics[59], which is very efficient in detecting phylogenetic signal[60,61]. Statistical significance of Abouheif's $C_{mean}$ was tested using 999 randomizations. As the phylogenetic signal was significantly present in all considered traits, we used phylogenetic eigenvector analysis[53,54] to account for species' non-independence and explore whether the observed differences between native and alien species can be ascribed to phylogenetic relationships among species. This method ordinates a double-centered matrix of phylogenetic proximities between each pair of species and produces a set of eigenvectors (hereafter referred to as phylogenetic eigenvectors), which can fully model the variation in trait values that is attributable to phylogeny. The matrix of phylogenetic proximities was calculated based on DaPhnE 1.0 dated phylogenetic supertree for northwestern and Central European angiosperms[62] using the Abouheif's proximity measure[59,60]. However, if all phylogenetic eigenvectors produced by this analysis are used as predictors in a multiple regression model, then all variability in the trait Y is explained, leaving no residual variation. It is therefore necessary to select only those eigenvectors that are able to model only the phylogenetic signal in trait Y. To select the optimal subset of phylogenetic eigenvectors, we used a procedure originally applied in the spatial context[63], termed the best performing selection method[64]. This procedure is based on an iterative search for the eigenvector that reduces the largest amount of phylogenetic autocorrelation in the residuals of trait Y. As new eigenvectors are added to the model, residuals are updated and autocorrelation re-estimated. The search stops when residual autocorrelation is reduced to a level that is statistically nonsignificant ($p > 0.05$). To measure and test the statistical significance of residual phylogenetic autocorrelation, we used Abouheif's $C_{mean}$ statistics[59] and 999 randomizations as described above. Species traits (i.e., SLA, plant height and seed weight) containing no phylogenetic signal were each obtained by extracting residuals from multiple linear regression of $\log_{10}$-transformed trait values and selected subsets of phylogenetic eigenvectors. These residuals were further analysed using the method described in section Single-trait analysis. We applied these analyses also on the dataset with imputed trait values. Phylogenetic eigenvector analysis was performed in the adephylo R package[65].

**Phylogenetic signal in the multiple-trait analysis**. For multivariate analyses in three-dimensional trait space, we also applied the above-mentioned method to select phylogenetic eigenvectors (derived from Abouheif's proximity matrix), but phylogenetic autocorrelation in multivariate data (i.e., in all three traits) was measured using the MULTISPATI method originally developed by Dray et al.[66] in a spatial context. This method utilizes a row-sum standardized weight matrix, here constructed using Abouheif's proximity, to exhibit autocorrelation in multivariate data, as measured by Moran's index[66,67]. The statistical significance of the multivariate autocorrelation index was calculated using 999 permutations of the rows of the table of residuals. This test is implemented in the multispati.randtest function of ade4 package[68]. A table of species traits containing no phylogenetic signal when ordinated in a multidimensional space was obtained by extracting residuals from multiple linear regression of $\log_{10}$-transformed and scaled trait table and a selected subset of phylogenetic eigenvectors. These residuals were further analysed using the method described in section Multiple-trait analysis. We applied these analyses also to the dataset with imputed trait values.

**Code availability**. All analyses were performed in R software (version 3.4.3)[69]. The R code used in this study is available at https://github.com/jdivisek/NativeVsAlienTraits.

## Data availability

A list of species occurring in each habitat type and maximum height for each species extracted from Kubát et al.[50] is available in Supplementary Data 1. SLA and seed weight data are available in the LEDA database[32] (https://uol.de/en/landeco/research/leda/). DaPhnE 1.0 dated phylogenetic supertree for northwestern and Central European angiosperms is available at https://doi.org/10.6084/m9.figshare.c.3305040.v1.

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

## Acknowledgements

J.D., M.C., and P.P. were supported by the Czech Science Foundation (Centre of Excellence Pladias, 14-36079G), Z.L. by the Czech Science Foundation (project 18-027738), P.P. by long-term research development project RVO 67985939 (The Czech Academy of Sciences), and D.M.R. by the DST-NRF Centre of Excellence for Invasion Biology, the National Research Foundation of South Africa (grant 85417) and the University of Vermont as a James Marsh Professor-at-Large. J.M. was supported by USDA NIFA 8062-2260-005-03S. B.B. acknowledges support from USDA NIFA project accession numbers 1009564 and 1014484. N.J.G. was supported by NSF-DEB 1257625

## Author contributions

J.M., M.C., P.P., D.M.R., N.J.G., and B.B. conceived the ideas. M.C. and Z.L. prepared the data. N.J.G., B.B., J.D., and Z.L. designed methodology. J.D. analyzed the data. J.D., M.C., and J.M. led the writing of the manuscript. P.P., D.M.R., N.J.G., B.B., and Z.L. contributed to the writing.

## Additional information

**Competing interests:** The authors declare no competing interests.

