## [Peer Review File · Nature Communications]

Reviewers' comments:

Reviewer #1 (Remarks to the Author):

This is a very interesting and well designed study. The novelty of the design in terms of grouping trait communities by habitat (rather than spatial areas) and separating 'invasive species' from 'naturalised species' is clever, and separates this work from other meta-analyses in this field. The findings are novel – naturalized plants have similar traits to natives but invasives are taller than natives. The paper is well written and the authors set the scene well by providing an overview of broader theoretical ecological concepts as well as the applied conservation importance.

With regard to the analyses presented in the paper the authors are to be commended for their novel approaches and for making their R code available for review. The permutation based approaches to model testing and the corrections to P-values for multiple testing were rigorous.

I have four major comments:

1. The classification of non-native species into the "naturalized" and "invasive" categories is critical as it is done in different ways depending on the author & purpose. While these authors specify an approach & cite the relevant papers it is vital that they briefly describe in this MS how the classification was arrived at and in the discussion they should mention how alternative classifications may affect (or not) the results. Catford et al 2016 may be a useful reference as they discuss how different traits may be associated with different dimensions of invasiveness & may impact on the correlation between traits & invasiveness. In particular I believe that the approach taken to classification to "invasive" takes only occurrences into account with widespread non-native species classed as "invasive". It may not be surprising therefore to see that plant height is a particularly diagnostic trait given its correlation with dispersal distance (e.g. Thomson et al 2011).

2. My second comment is related to the first. Given a particular definition of "invasive" it would be very interesting to compare the "invasives" with a subset of natives that also show the same demographic/occurrence patterns. Would a comparison between invasive & widespread native species still show a difference in trait values? Is the proportion of non-native species with the correlated demographic and functional traits much larger than the proportion of native species with the correlated demographic and functional traits?

3. I would like to see further justification for the author's decision to use CDF's in their analysis of the dispersion of traits values in multivariate space. It was not clear why this step of constructing and comparing CDF's was necessary. Specifically, my understanding is that the authors measured the distance from each species in multivariate space to the centroid of the multivariate space representing the native species community. Why were these distance measures not directly compared (i.e. the distances of the 'invasive species from the centroid' as a list of values vs. the distances of the 'native species from the centroid')? It seems that by constructing the CDF's the authors introduced an issue described in line 349, which was that 'If CDFs crossed each other, we considered the larger area only, i.e. that below (negative values) or above (positive values) the CDF for native species' (i.e. only part of the value distribution was compared). Furthermore, readers may find the CDF's less easy to interpret than raw distances. This is a pedantic point, and I acknowledge that a direct comparison of the distances is unlikely to change the results, but I would like to know why the authors chose this approach (preferably with a citation) or see the analysis done with the raw distance values.

4. With regard to the interpretation of the results, the authors present the results of analysis of both the raw data and the same models after accounting for the phylogenetic structure of the species data. The latter of these analyses shows no significant differences in multivariate space between either 'invasive vs. native' or 'native vs. naturalised' comparisons. Given that accounting for the phylogenetic relatedness of species is now the norm in comparative studies of this kind, I think that the authors understate the importance of this result i.e. that there is no statistically significant difference between the locations of the invasive and native species in multivariate trait space (and conversely overstate the importance of the non-phylogenetically corrected result). For example, it appears to me that there is not enough evidence to support the assertion in the discussion that 'invasive species differ from both native and naturalized non-invasive species by occupying the periphery of the plant functional trait space represented in each habitat'. Certainly,

there is a strong trend in this direction but this could be driven by the fact that the invasive species are more similar to each other due to their shared phylogenetic history (e.g. if the invasive plants tend to come from the same place). I think this is an issue that the authors could easily fix with some minor reframing of the discussion. The discussion already states on a couple of occasions that differences in traits are largely driven by the difference in height which is observed between species when compared at a habitat level. This difference is robust to phylogeny. In combination, with the absence of an a difference between naturalised and native species in most habitats, and the absence of differences in two the other two traits (SLA and seed weight) between native and invasive species this still supports the authors hypothesis that functional similarity is important for naturalisation whilst differentiation in competitive ability (i.e. for light in this case) is an important trait for successful invasion.

Minor points:

Could time since introduction be used as an explanatory variable? It is likely to be important for assessing the invasiveness criteria.

Line 106 - Please explain r-K continuum very briefly for a general readership. Nature communications will be read by non-ecologists.

Line 126 – If there were no significant differences in SLA, please state this result also here.

Line 222 – Given that there is no evidence of differences in SLA in the results, should this read 'not by SLA' rather than 'hardly by SLA'

Line 354 – I assume that the CDFs were recalculated after permuting the data. Please state this here.

Line 605 – In Figure legend 2, i.e. is repeated.

Reviewer #2 (Remarks to the Author):

This paper evaluates the role of functional trait distances in determining plant invasiveness. This is a hot topic within the invasion biology literature and many experimental observational and meta-analytical studies have been conducted to try to answer the question: What is the functional profile that makes invasive an exotic species?

Using three functional traits (height, seed mass and specific leaf area), the authors mainly found that the amount of trait distances distinguish the invasiveness degree of exotic plant species. If exotic species are naturalized but not invasive, their functional profile is similar to that shown by native species. However, if exotic plant species are invasive their functional profile is significantly different compared to native communities. Therefore, the main message of this contribution is that exotic species have to be somehow similar to the native community in order to establish but simultaneously they will become invasive only when they show significant functional trait differences.

One of the main strength of the paper is that the main findings are consistent across different habitats and for both univariate (mainly height) and multivariate regressions. Another strength is broad the spatial scale addressed here thanks to using the unique database of the Czech phytosociological records.

Overall, I think the paper has enough novelty to be published in a high-ranked journal. Many previous studies have not considered different types of invasiveness degree (naturalized and invasive in this case) under such broad scale when testing trait differences between exotic and native communities. Thanks to this approach the authors are able to show a differential pattern of trait differences between native and exotics depending on the degree of invasiveness that is consistent across very different habitats. Moreover and although the authors do not say explicitly in the manuscript, this work shows for the first time a non-linear relationship between functional

trait differences and the degree of invasiveness. I am referring to Fig. 2, which in my opinion is a brilliant presentation of the main findings.

Yet, I think the paper needs a strong revision to better highlight the novelty of the contribution, to better show the caveats of the study, and to better present and justify some of the methods used.

Major comments

The paper clearly presents that exotic naturalized species are statistically similar to native communities but invasive species are not. I think it will be very interesting to know at which functional trait distance (either height or multitrait distance) occurs this transition from naturalized to invasive. For instance, in lines 123-124 it is presented that naturalized species are far apart from native communities a distance of 0.2-0.3 meters on average but invasive species where 1.2 taller on average. This is ok, but I think it would be more interesting to present 1) the range of trait distance that makes the transition from naturalized to invasive species (that has to be some distance between 0.2 and 1.2 meters according to the results), 2) which is the limit of the differences (it has to be some limit, otherwise the paper does not support the idea that exotic species need to be similar and different at the same time in order to become invasive) and 3) whether the transition range and the limit of trait differences is similar across habitats. That would make the paper more interesting than rather presenting dichotomous results of statistical differences between native-naturalized and native-invasive. That would also help to show the tipping point of the non-linear relationship that determine the degree of invasiveness (naturalized and invasive), giving more weight in the paper to Fig. 2, that in my opinion is the main interesting result.

Moreover, the paper presents the idea according to the statistical regression differences that being different is synonymous of being invasive. However, this test lacks an important component, and these are species that were exotic but have fail to naturalized and invade. It is likely that many of these species were significant different in their functional profile with respect to the native community. I understand that this information is always very hard to obtain, yet the authors could solve this problem by presenting clearer whether there is a specific direction of the significant functional trait differences. According to the results, taller is the specific direction determining invasiveness, less clear is the pattern in multivariate space. The authors state in lines 210-213 that being in the periphery is important but I wonder whether there is a specific region in the periphery promoting the invasiveness potential or according to their results all locations of the periphery are equally important. I would guess that the answer is no, and there is a specific region in the periphery that accounts for most of the invasive species. There has to be a combination of height, SLA and seed mass that promote exotic species to be invasive. If not, there would be contradicting results between what was found for height (taller is always better to become invasive) and what was found for multivariate analyses.

Lines 319-323. Trait imputation is an important issue, even more here where almost one third of the species did not have information of SLA and seed mass, but little details are given. The authors need to give more information about the methodology used for trait imputation and which were the stress values. These stress values gives an idea of how parsimonious were the values assigned to these species with NA.

Lines 319-323 Another important issue regarding the trait imputation is that if phylogenies are used for trait imputation (as the authors say in lines 321-322, when the imputed trait values using average for genera), then values obtained from statistical models testing trait differences can not use information of species relatedness because the trait imputation process tend to increase the phylogenetic trait signal. In the case of this paper, trait imputation needs to be better addressed with random forest functions (see Penone et al. MEE)

Minor comments

It is surprising that among all methods for estimating phylogenetic signal the authors only considered Abouheif's C_{mean} . I think the authors need to also present other indices such as Pagel Lambda or Bolmberg K and if not justify in the paper why they only used Abouheif metric.

Overall, this paper is a strong contribution helping to better understanding the causes by which only a small fraction of exotic species become invasive. These results have also importance to design strategies for the control and management of exotic plant species in natural ecosystems. I hope my comments help the authors to come up better quality version of their manuscript.

Best regards,
Oscar Godoy.

Reviewers' comments:

Reviewer #1 (Remarks to the Author):

This is a very interesting and well designed study. The novelty of the design in terms of grouping trait communities by habitat (rather than spatial areas) and separating 'invasive species' from 'naturalised species' is clever, and separates this work from other meta-analyses in this field. The findings are novel – naturalized plants have similar traits to natives but invasives are taller than natives. The paper is well written and the authors set the scene well by providing an overview of broader theoretical ecological concepts as well as the applied conservation importance.

With regard to the analyses presented in the paper the authors are to be commended for their novel approaches and for making their R code available for review. The permutation based approaches to model testing and the corrections to P-values for multiple testing were rigorous.

I have four major comments:

1. The classification of non-native species into the “naturalized” and “invasive” categories is critical as it is done in different ways depending on the author & purpose. While these authors specify an approach & cite the relevant papers it is vital that they briefly describe in this MS how the classification was arrived at and in the discussion they should mention how alternative classifications may affect (or not) the results. Catford et al 2016 may be a useful reference as they discuss how different traits may be associated with different dimensions of invasiveness & may impact on the correlation between traits & invasiveness. In particular I believe that the approach taken to classification to “invasive” takes only occurrences into account with widespread non-native species classed as “invasive”. It may not be surprising therefore to see that plant height is a particularly diagnostic trait given its correlation with dispersal distance (e.g. Thomson et al 2011).

We expanded respective parts of the text to address these points (l. 329-350). The concept we used to classify alien species of the Czech Republic is based on unified framework for biological invasions proposed by Richardson et al. (2000, 2011), Richardson & Pyšek (2006), Blackburn et al. (2011) that describes how species proceed in the invasion process by overcoming geographical, environmental and biotic barriers. In the revised manuscript we provide detailed definitions of “naturalized” and “invasive” categories. Throughout the paper we use standard terminology that is widely accepted in the invasion literature and feel that it is beyond the scope of this paper to discuss the concepts and frameworks in detail.

2. My second comment is related to the first. Given a particular definition of “invasive” it would be very interesting to compare the “invasives” with a subset of natives that also show the same demographic/occurrence patterns. Would a comparison between invasive & widespread native species still show a difference in trait values? Is the proportion of non-native species with the correlated demographic and functional traits much larger than the proportion of native species with the correlated demographic and functional traits?

Thank you for this comment. This is a very intriguing idea but we believe the analysis is beyond the scope of the current study, because we do not study demography, but we are asking how the differences in traits translate into invasiveness. To do it properly would require that we construct rank abundance curves for each of the habitats and compare native and invasive species of similar rank. However, we feel that the degree to which native species would be considered demographically similar to alien/invasive species, would introduce a great deal of subjectivity into the paper.

3. I would like to see further justification for the author's decision to use CDF's in their analysis of the dispersion of traits values in multivariate space. It was not clear why this step of constructing and comparing CDF's was necessary. Specifically, my understanding is that the authors measured the distance from each species in multivariate space to the centroid of the multivariate space representing the native species community. Why were these distance measures not directly compared (i.e. the distances of the 'invasive species from the centroid' as a list of values vs. the distances of the 'native species from the centroid')? It seems that by constructing the CDF's the authors introduced an issue described in line 349, which was that 'If CDFs crossed each other, we considered the larger area only, i.e. that below (negative values) or above (positive values) the CDF for native species' (i.e. only part of the value distribution was compared).

Furthermore, readers may find the CDF's less easy to interpret than raw distances. This is a pedantic point, and I acknowledge that a direct comparison of the distances is unlikely to change the results, but I would like to know why the authors chose this approach (preferably with a citation) or see the analysis done with the raw distance values.

For these kinds of data, a parametric t-test could be used to compare the average distance of each species group (native, naturalized and invasive) from the centroid of native species. However, this kind of test compares only the means of the two groups, whereas there could be differences in the variance, skewness, or other moments of the distribution. Parametric tests are available for higher moments, but these tests are increasingly sensitive to outliers and influential points. Therefore, we used a more general test of the difference in the cumulative distribution functions (CDF) for native species and naturalized or invasive species. A standard Kolmogorov-Smirnov CDF test is based on the maximum difference between two curves, and a look-up table is used to estimate a tail probability for CDF differences based on ranked observations. For our test, we thus calculated the summed area difference between the CDF for native species and the CDF for either naturalized or invasive species (see also Cayuela et al. (2015) for a similar approach to comparing rarefaction curves). We added this justification to the text (l. 408-430).

However, to check the soundness of the overall approach, we also plotted, for each species group, the distribution of distances away from the native group centroid. These violin plots are included in Supplementary Materials (Supplementary Figures 3, 6, 12 and 16). Then we calculated nonparametric Wilcoxon Rank Sum test to test shifts in average distances of naturalized or invasive species (please see Tables 1 and 2 below). Results of these comparisons did not contradict previous ones based on CDFs. As it is visible from the violin plots, invasive species are in all cases further from the centroid whereas naturalized species are distributed similarly to native species.

Table 1 – Wilcoxon Rank Sum tests of the shifts in average distances of naturalized or invasive species from the centroid of native species in the 3D trait space. Species with missing value of any of the three traits were removed.

	Naturalized species			Invasive species		
	W	P	P _{adj.}	W	P	P _{adj.}
Observed species traits						
Grassland and heathland vegetation	38170	0.009	0.053	10082	0.012	0.014
Ruderal and weed vegetation	41848	0.046	0.130	10186	0.004	0.012
Rock and scree vegetation	6966	0.283	0.339	1783	0.011	0.014
Wetland vegetation	15741	0.065	0.130	5461	0.014	0.014
Scrub vegetation	24363	0.768	0.768	5663	0.011	0.014
Forest vegetation	23173	0.092	0.138	9525	0.003	0.012
Residuals of phylogenetic models						
Grassland and heathland vegetation	42092	0.299	0.749	9740	0.032	0.069
Ruderal and weed vegetation	45849	0.735	0.749	10218	0.003	0.021
Rock and scree vegetation	7448	0.747	0.749	1602	0.076	0.076
Wetland vegetation	16468	0.204	0.749	5262	0.035	0.069
Scrub vegetation	23858	0.545	0.749	5221	0.069	0.076
Forest vegetation	25544	0.749	0.749	8559	0.066	0.076

Table 2 – Wilcoxon Rank Sum tests of the shifts in average distances of naturalized or invasive species from the centroid of native species in the 3D trait space. Missing trait values were imputed based on correlations among traits and species' phylogenetic relatedness.

	Naturalized species			Invasive species		
	W	P	P _{adj.}	W	P	P _{adj.}
Observed species traits						
Grassland and heathland vegetation	86483	0.157	0.315	23331	0.000	0.001
Ruderal and weed vegetation	85466	0.033	0.101	22448	0.000	0.000
Rock and scree vegetation	11302	0.661	0.681	2741	0.011	0.011
Wetland vegetation	27894	0.034	0.101	11901	0.003	0.003
Scrub vegetation	41158	0.681	0.681	13007	0.001	0.001
Forest vegetation	47237	0.593	0.681	26091	0.000	0.000
Residuals of phylogenetic models						
Grassland and heathland vegetation	93901	0.822	0.983	20993	0.023	0.046
Ruderal and weed vegetation	95395	0.769	0.983	20764	0.005	0.030
Rock and scree vegetation	11676	0.983	0.983	2283	0.244	0.244
Wetland vegetation	32030	0.965	0.983	10509	0.106	0.159
Scrub vegetation	41485	0.780	0.983	10937	0.138	0.166
Forest vegetation	52185	0.250	0.983	22411	0.015	0.045

4. With regard to the interpretation of the results, the authors present the results of analysis of both the raw data and the same models after accounting for the phylogenetic structure of the species data. The latter of these analyses shows no significant differences in multivariate space between either 'invasive vs. native' or 'native vs. naturalised' comparisons. Given that accounting for the phylogenetic relatedness of species is now the norm in comparative studies of this kind, I think that the authors understate the importance of this result i.e. that there is no statistically significant difference between the locations of the invasive and native species in multivariate trait space (and conversely overstate the importance of the non-phylogenetically corrected result). For example, it appears to me that there is not enough evidence to support the assertion in the discussion that 'invasive species differ from both native and naturalized non-invasive species by occupying the periphery of the plant

functional trait space represented in each habitat'. Certainly, there is a strong trend in this direction but this could be driven by the fact that the invasive species are more similar to each other due to their shared phylogenetic history (e.g. if the invasive plants tend to come from the same place). I think this is an issue that the authors could easily fix with some minor reframing of the discussion. The discussion already states on a couple of occasions that differences in traits are largely driven by the difference in height which is observed between species when compared at a habitat level. This difference is robust to phylogeny. In combination, with the absence of an a difference between naturalised and native species in most habitats, and the absence of differences in two the other two traits (SLA and seed weight) between native and invasive species this still supports the authors hypothesis that functional similarity is important for naturalisation whilst differentiation in competitive ability (i.e. for light in this case) is an important trait for successful invasion.

While revising the manuscript we paid more attention to phylogenetically corrected analyses. We have improved the description of phylogenetic background of our hypotheses in the Introduction (l. 58–65) a reframed discussion related to the results of these analyses (l. 278–299).

Minor points:

Could time since introduction be used as an explanatory variable? It is likely to be important for assessing the invasiveness criteria.

Unfortunately, information on time when each particular plot or habitat has been first invaded is not available for this kind of data. However, it has been repeatedly shown in the invasion literature that time since introduction is closely and positively related to regional distribution and also to the probability that the species becomes invasive – this is analysed for our target region in Pyšek & Jarošík (2005). Residence time determines the distribution of alien plants. In: Inderjit (ed.), Invasive plants: ecological and agricultural aspects, p. 77–96, Birkhäuser Verlag-AG, Basel. Therefore, the effect of the time since introduction is actually manifested in the separation of naturalized/invasive in a way.

Line 106 - Please explain r-K continuum very briefly for a general readership. Nature communications will be read by non-ecologists.

Explanation was added (l. 108-111)

Line 126 – If there were no significant differences in SLA, please state this result also here.

Corrected

Line 222 – Given that there is no evidence of differences in SLA in the results, should this read 'not by SLA' rather than 'hardly by SLA'

Corrected

Line 354 – I assume that the CDFs were recalculated after permuting the data. Please state this here.

Corrected

Line 605 – In Figure legend 2, i.e. is repeated.

Corrected

Reviewer #2 (Remarks to the Author):

This paper evaluates the role of functional trait distances in determining plant invasiveness. This is a hot topic within the invasion biology literature and many experimental observational and meta-analytical studies have been conducted to try to answer the question: What is the functional profile that makes invasive an exotic species?

Using three functional traits (height, seed mass and specific leaf area), the authors mainly found that the amount of trait distances distinguish the invasiveness degree of exotic plant species. If exotic species are naturalized but not invasive, their functional profile is similar to that shown by native species. However, if exotic plant species are invasive their functional profile is significantly different compared to native communities. Therefore, the main message of this contribution is that exotic species have to be somehow similar to the native community in order to establish but simultaneously they will become invasive only when they show significant functional trait differences.

One of the main strength of the paper is that the main findings are consistent across different habitats and for both univariate (mainly height) and multivariate regressions. Another strength is broad the spatial scale addressed here thanks to using the unique database of the Czech phytosociological records.

Overall, I think the paper has enough novelty to be published in a high-ranked journal. Many previous studies have not considered different types of invasiveness degree (naturalized and invasive in this case) under such broad scale when testing trait differences between exotic and native communities. Thanks to this approach the authors are able to show a differential pattern of trait differences between native and exotics depending on the degree of invasiveness that is consistent across very different habitats. Moreover and although the authors do not say explicitly in the manuscript, this work shows for the first time a non-linear relationship between functional trait differences and the degree of invasiveness. I am referring to Fig. 2, which in my opinion is a brilliant presentation of the main findings.

Yet, I think the paper needs a strong revision to better highlight the novelty of the contribution, to better show the caveats of the study, and to better present and justify some of the methods used.

Major comments

The paper clearly presents that exotic naturalized species are statistically similar to native communities but invasive species are not. I think it will be very interesting to know at which

functional trait distance (either height or multitrait distance) occurs this transition from naturalized to invasive. For instance, in lines 123-124 it is presented that naturalized species are far apart from native communities a distance of 0.2-0.3 meters on average but invasive species where 1.2 taller on average. This is ok, but I think it would be more interesting to present 1) the range of trait distance that makes the transition from naturalized to invasive species (that has to be some distance between 0.2 and 1.2 meters according to the results), 2) which is the limit of the differences (it has to be some limit, otherwise the paper does not support the idea that exotic species need to be similar and different at the same time in order to become invasive) and 3) whether the transition range and the limit of trait differences is similar across habitats. That would make the paper more interesting than rather presenting dichotomous results of statistical differences between native-naturalized and native-invasive. That would also help to show the tipping point of the non-linear relationship that determine the degree of invasiveness (naturalized and invasive), giving more weight in the paper to Fig. 2, that in my opinion is the main interesting result.

Thank you for this comment. For each species group in each habitat type, we have now plotted the distribution of distances away from native group centroid using violin plots to clearly show the differences between naturalized and invasive species. These plots are included in the Supplementary Materials (Supplementary Figures 3, 6, 12 and 16). If we consider an overlap of interquartile ranges for naturalized and invasive species as a transitional zone, then the transition between naturalized and invasive species occurs somewhere between the distances of 1 and 1.5, i.e. 20-30% of the most distant species. This zone seems to be relatively consistent across all considered habitats. However, this is an arbitrary delineation of a transitional zone; in fact, the transition between naturalized and invasive species is rather blurred.

Regarding the limit of the differences: In terms of the limiting similarity hypothesis, any species new to the habitat would need to be at least as distant as the closest invasive species to become invasive, i.e. the limit would be the distance of the closest invasive species, which is already present in the habitat. On the other hand, in terms of the environmental filtering hypothesis, any species new to the habitat would need to be closer or as far as the most distant species in the trait space. Otherwise, the species would not be able to establish in the habitat.

To address these points, we added a second x axis with original distance units (Std. dev. units) to CDF plots (to show different upper limits in the last two habitats) and plotted species distances using violin plots. However, we still prefer not to show any exact numbers about limits and transition zones, because they would be very speculative. Instead, we rather show average distances of native, naturalized and invasive species and their differences (l. 157-168).

Moreover, the paper presents the idea according to the statistical regression differences that being different is synonymous of being invasive. However, this test lacks an important component, and these are species that were exotic but have fail to naturalized and invade. It is likely that many of these species were significant different in their functional profile with respect to the native community. I understand that this information is always very hard to obtain, yet the authors could solve this problem by presenting clearer whether there is a

specific direction of the significant functional trait differences. According to the results, taller is the specific direction determining invasiveness, less clear is the pattern in multivariate space. The authors state in lines 210-213 that being in the periphery is important but I wonder whether there is a specific region in the periphery promoting the invasiveness potential or according to their results all locations of the periphery are equally important. I would guess that the answer is no, and there is a specific region in the periphery that accounts for most of the invasive species. There has to be a combination of height, SLA and seed mass that promote exotic species to be invasive. If not, there would be contradicting results between what was found for height (taller is always better to become invasive) and what was found for multivariate analyses.

We followed your suggestion and, for each habitat type, we have now quantified the number of species in each of the eight regions (octants; Figure 2) of the trait space (Figure 3 and Supplementary Figures 5, 11 and 15). There is obvious tendency of invasive species towards above-average height and seed weight. We added this result to the paper (l. 145–156) and the numbers of species per octant are available in Supplementary Materials (Supplementary Tables 3, 4, 7 and 8).

Lines 319-323. Trait imputation is an important issue, even more here where almost one third of the species did not have information of SLA and seed mass, but little details are given. The authors need to give more information about the methodology used for trait imputation and which were the stress values. These stress values gives an idea of how parsimonious were the values assigned to these species with NA.

Lines 319-323 Another important issue regarding the trait imputation is that if phylogenies are used for trait imputation (as the authors say in lines 321-322, when the imputed trait values using average for genera), then values obtained from statistical models testing trait differences can not use information of species relatedness because the trait imputation process tend to increase the phylogenetic trait signal. In the case of this paper, trait imputation needs to be better addressed with random forest functions (see Penone et al. MEE)

Trait imputation is always an important issue because different approaches may give very different results. In our study, we preferred the simplest way of imputing missing trait values, which was based on simple averaging of trait values across genera or families (in cases when trait values of all species of the genus were missing). Now, we have also applied the missForest function and imputed missing trait values based on correlations among traits as you have suggested. However, we are convinced that this is not appropriate approach for our data because we have only three traits, and about 30% of values are missing in two of them. Therefore, we prefer to follow the recommendation of Penone et al. (2014) who concluded that “adding phylogenetic information into the imputation algorithms improves estimation of missing values” and used first 10 phylogenetic eigenvectors as additional explanatory variables in missForest. Result of this imputation was very similar to that we obtained with our original approach (for SLA $r = 0.73$ and for Seed weight $r = 0.79$). In the revised manuscript, we described all methods we used for trait imputation (l. 364-378), but we present only the results based on the last approach using missForest with phylogenetic eigenvectors.

In terms of phylogenetic signal in imputed trait data: Trait imputation approaches using phylogenies indeed increase phylogenetic signal, but we are convinced that this is not problem at all in our study. We did not use information on species relatedness directly in the models testing trait differences. Instead, we first established models where selected phylogenetic eigenvectors were used as explanatory variables (the number of selected eigenvectors was dependent on the strength of phylogenetic signal) and then we extracted residuals of these models to obtain trait values, which do not exhibit statistically significant phylogenetic signal. These residuals were subsequently analysed to test trait differences between species groups. In these models, we thus tested differences in species traits after accounting phylogenetic signal, but the models did not use information on species relatedness.

Minor comments

It is surprising that among all methods for estimating phylogenetic signal the authors only considered Abouheif's C_{mean} . I think the authors need to also present other indices such as Pagel Lambda or Bolmberg K and if not justify in the paper why they only used Abouheif metric.

We followed recommendation of Münkemüller et al. (2012) in MEE who showed that Abouheif's C_{mean} and Pagel's lambda perform well and substantially better than Blomberg's K. However, Pagel's lambda is computationally very demanding and that is why we decided to use Abouheif's C_{mean} .

Overall, this paper is a strong contribution helping to better understanding the causes by which only a small fraction of exotic species become invasive. These results have also importance to design strategies for the control and management of exotic plant species in natural ecosystems. I hope my comments help the authors to come up better quality version of their manuscript.

Thank you for your comments and for considering our paper an important contribution.

Best regards,
Oscar Godoy.

REVIEWERS' COMMENTS:

Reviewer #1 (Remarks to the Author):

The revised MS has addressed my concerns adequately. I have no further comments.

Reviewer #2 (Remarks to the Author):

Dear Editor,

I do not have more comments to add. The authors have correctly addressed all my previous comments. I think the revised version presents more solid and strong arguments of the effect of trait differences in determining invasion success. Particularly, I appreciate the effort done in the paper to present the multivariate approach showing which is the trait space occupied by invasive species and how this trait space differ between this group and the groups of naturalized and native species.

I do hope this contribution makes a significant impact in the literature.

Best regards,
Oscar Godoy